# Effects of Clinical Wastewater on the Bacterial Community Structure from Sewage to the Environment

**DOI:** 10.3390/microorganisms9040718

**Published:** 2021-03-31

**Authors:** Ilse Verburg, H. Pieter J. van Veelen, Karola Waar, John W. A. Rossen, Alex W. Friedrich, Lucia Hernández Leal, Silvia García-Cobos, Heike Schmitt

**Affiliations:** 1Wetsus, European Centre of Excellence for Sustainable Water Technology, 8900 CC Leeuwarden, The Netherlands; ilse.verburg@wetsus.nl (I.V.); pieter.vanveelen@wetsus.nl (H.P.J.v.V.); lucia.hernandez@wetsus.nl (L.H.L.); 2Department of Medical Microbiology and Infection Prevention, University Medical Center Groningen, University of Groningen, 9713 GZ Groningen, The Netherlands; j.w.a.rossen@rug.nl (J.W.A.R.); alex.friedrich@umcg.nl (A.W.F.); s.garciacobos@isciii.es (S.G.-C.); 3Izore, Centrum Infectieziekten Friesland, 8900 JA Leeuwarden, The Netherlands; k.waar@izore.nl; 4Institute for Risk Assessment Sciences, Utrecht University, 3508 TD Utrecht, The Netherlands; 5Centre for Infectious Disease Control, National Institute for Public Health and the Environment (RIVM), 3721 MA Bilthoven, The Netherlands

**Keywords:** 16S rRNA amplicon sequencing, bacterial community structure, clinical wastewater, sampling campaign, wastewater pathway

## Abstract

This study pertains to measure differences in bacterial communities along the wastewater pathway, from sewage sources through the environment. Our main focus was on taxa which include pathogenic genera, and genera harboring antibiotic resistance (henceforth referred to as “target taxa”). Our objective was to measure the relative abundance of these taxa in clinical wastewaters compared to non-clinical wastewaters, and to investigate what changes can be detected along the wastewater pathway. The study entailed a monthly sampling campaign along a wastewater pathway, and taxa identification through 16S rRNA amplicon sequencing. Results indicated that clinical and non-clinical wastewaters differed in their overall bacterial composition, but that target taxa were not enriched in clinical wastewater. This suggests that treatment of clinical wastewater before release into the wastewater system would only remove a minor part of the potential total pathogen load in wastewater treatment plants. Additional findings were that the relative abundance of most target taxa was decreased after wastewater treatment, yet all investigated taxa were detected in 68% of the treated effluent samples—meaning that these bacteria are continuously released into the receiving surface water. Temporal variation was only observed for specific taxa in surface water, but not in wastewater samples.

## 1. Introduction

Antimicrobial resistance (AMR) is recognized as a major threat to public health at a global scale [1]. The ESKAPE pathogens (*Enterococcus faecium*, *Staphylococcus aureus*, *Klebsiella pneumoniae*, *Acinetobacter baumannii*, *Pseudomonas aeruginosa*, and *Enterobacter* species) play an important role in nosocomial infections, pathogenesis, and AMR spread [2,3]. In 2018 the World Health Organization (WHO) published a global priority list of antimicrobial-resistant bacteria (AMRB) for which research and development of new antibiotics is urgently needed [4]. 

After consumption of antibiotics, the bacterial composition in the gut can be altered [5,6] and even be enriched in AMRB [7]. Some bacteria are found to thrive after antibiotic treatment and can cause antibiotic-associated diarrhoea (AAD) [8,9,10]. Such, and other, pathogenic bacteria are expected to be more abundant in clinical settings, where the consumption of antibiotics is higher than in the general community [11]. Previous research showed that clinical wastewaters contain a higher level of AMR bacteria, antimicrobial resistance genes and antimicrobial residues than non-clinical wastewaters [12,13,14,15]. Nonetheless, clinical wastewater constitutes a minor proportion of all wastewater and the magnitude of the impact that clinical wastewater has on AMR bacterial load in downstream wastewaters is not yet fully understood. Paulus et al. showed that pre-treatment of hospital wastewater helps to reduce the presence of ARGs in the receiving WWTP [13], yet Buelow et al. did not observe differences in the relative abundance of ARGs in WWTPs that did or did not receive hospital wastewater [14]. More insight into the bacterial composition of wastewater is critical to decide on the benefit of dedicated wastewater treatment at the level of clinical institutions. 

Human gut bacteria are released into the wastewater system via feces and can reach the environment via this pathway. Wastewater from both clinical and non-clinical settings converges in the wastewater treatment plant (WWTP). WWTP’s are designed to reduce the biological oxygen demand, nitrogen, phosphorus, and the total suspended solids, but not the removal of pharmaceutically active compounds and bacteria. Earlier studies have shown that most pathogenic bacteria are decreased in their relative abundance in the WWTP, while others (e.g., *Mycobacterium* spp. and *Clostridium* spp.) increase [16,17]. Although wastewater and surface waters are different environments from the human gut, several pathogenic bacteria are known to be water-borne pathogens (species that are able to spread via aquatic ecosystems) while others can be classified as water-based pathogens (species that are able to grow and thrive in water systems) [18,19,20,21,22]. 

This study aimed to investigate the contribution of clinical wastewater on the bacterial composition in wastewater and the environment. We used 16S rRNA to broadly screen the relative abundance and fate of bacterial target taxa that were selected based on pathogenic potential: (i) pathogens of clinical relevance and AMR features (ESKAPE pathogens and WHO priority pathogens) [2,3,4]; (ii) water-based pathogens that can grow and thrive in water systems [18,19,20,21,22]; (iii) antibiotic-associated diarrhea (AAD) bacteria [8,9,10] and; (iv) bacteria that increase after WWTP treatment [16,17].

The results of this study provide insight into the community composition origin in a wastewater chain, the difference between clinical and non-clinical wastewater, and how this affects the bacterial community in the WWTP and the receiving surface water. In doing so, the study sheds light on the likely fate of potentially pathogenic bacteria in this wastewater chain.

## 2. Materials and Methods

### 2.1. Sampling Campaign

The sampling campaign was conducted in 2017 across the wastewater network in Sneek (33,855 inhabitants), The Netherlands. While a more detailed description of the sampling campaign can be found in our previous research [12] a brief summary of the study is as follows: wastewater sample locations encompassed: (a) hospital (300 beds), nursing home (220 beds) and domestic (80 households) wastewater sources (all sites selected to exclude the influence of industrial waste and/or rainwater); (b) influent and effluent from the conventional WWTP at Sneek (aerobic treatment, 73,000 p. e.) which receives wastewater from the hospital, nursing home and city district where the domestic wastewater sample was obtained. Wastewater samples were collected as twenty-four-hour samples sampled flow proportionally (WWTP) or time proportionally (domestic, hospital, and nursing home wastewater). Surface water samples were collected as grab samples, taken at ~20 cm depth and one-meter distance from the waterside. Surface water samples were taken from the receiving surface water of the Geeuw canal at two locations, 330 m south-west (N 53°02′15.10″, E 5°63′72.76″) and 388 m north-east (N 53°02′72.15″, E 5°64′28.97″) from the WWTP discharge point (N 53°02′38.85″, E 5°64′03.20″), and from a nature reserve “de Deelen” which has previously been shown to lack human influence [23]. The locations of the sampling points in Sneek are indicated in the map of Sneek in Appendix A. Samples collected twice per month for one year, transported cooled and processed within the same day.

### 2.2. DNA Extraction and 16S rRNA Gene Amplicon Sequencing

Water samples were filtered in volumes of 25 mL (wastewaters) or 200 mL (surface water and effluent) using sterile 0.22 µm polyvinylidene difluoride (PVDF)-membrane filters. Filters were stored at −80 °C until DNA extraction was performed. DNA was extracted using the DNeasy Power Water kit (Qiagen, Hilden, Germany) and quantified using the Qubit dsDNA BR (broad range) Assay kit (ThermoFisher Scientific, Waltham, MA, USA) according to the manufacturer’s instructions. DNA was stored at −20 °C before subsequent analysis. Amplicon sequencing of the V3–V4 regions of the 16S rRNA gene was performed on an Illumina MiSeq (Illumina, San Diego, CA, USA). Libraries were prepared by using the Nextera XT DNA Library Preparation Kit following the 16S Metagenomic Sequencing Library Preparation protocol, according to the manufacturer’s instructions [24]: the V3-V4 regions of the 16S rRNA gene were amplified by the polymerase chain reaction (PCR) using Amplicon primers with overhang adaptors (16S Amplicon PCR Forward Primer = 5′ TCGTCGGCAGCGTCAGATGTGTATAAGAGACAGCCTACGGGNGGCWGCAG 16S Amplicon PCR Reverse Primer = 5′ GTCTCGTGGGCTCGGAGATGTGTATAAGAGACAGGACTACHVGGGTATCTAATCC). A second PCR was used for attaching indices and Illumina sequencing adapters using the Nextera^®^ XT Index Kit (Illumina, San Diego, CA, USA). Fragments were cleaned after each PCR using freshly prepared Ampure XP Beads (Beckman Coulter Genomics, Danvers, MA, USA). In total, 171 samples were successfully sequenced, resulting in an average of 50,734 reads per sample (Table 1). The sample with the lowest number of reads was a downstream sample (18,059 reads), and the sample with the highest number of reads came from the hospital (138,846 reads). Sequence data are available in the NCBI sequence read archive (SRA) under project numbers PRJNA668059 and PRJNA668064.

### 2.3. Data Processing and Visualization

Adaptor sequences were removed from the FASTQ and reads were filtered by length (200–550 nucleotides) using Qiagen CLC Bio Genomics Workbench 10.1.1. (Qiagen, Germantown, MD, USA). Sequence reads were de-noised using DADA2 (v. 1.11.0) in R Statistical Software (v. 3.5.0). Taxonomy was assigned to representative sequences for amplicon sequence variants (ASV) using the SILVA database (v. 132) [25]. The sample by ASV frequency matrix was combined with taxonomic assignments using the biom-format package. Representative sequences were then imported into QIIME2 (v. 2018.11) [26] for alignment using MAFFT [27]. The alignment was then filtered for gaps and used to build a phylogeny using Fasttree2 [28]. These data were then imported in R as a phyloseq object [29] and filtered to exclude Chloroplasts and Mitochondria and to retain only Bacteria for downstream analyses of microbial diversity and community structure.

ASV counts were normalized to the total number of reads per sample for calculating unweighted and weighted UniFrac distances among samples [30,31]. Principal coordinates analysis (PCoA) visualizations based on the distance matrices were used to assess the phylogenetic composition and structure of bacterial communities at different locations along the wastewater pathway. Global and pairwise PERMANOVA [32,33] were applied to test for differential clustering of sample locations. Additional tests of homogeneity of group dispersion were performed with the betadisper function and a permutation test with 999 permutations.

#### 2.3.1. Selection of Target Taxa for Fate Monitoring through Wastewater Pathway

A total of 24 bacterial genera or species were selected as “target taxa” to study their changes in relative abundance and fate along the studied wastewater pathway (Table 2), based on pathogenicity, antibiotic resistance potential and known or possible association with WWTP. The taxa selection included: (1) the ESKAPE pathogens that are of particular clinical relevance and prone to antibiotic resistance [2,3], (2) the WHO priority pathogens for AMR [4], (3) water-based pathogens [18,19,20,21,22], (4) bacteria which are AAD associated/increased after antibiotic (AB) treatment [8,9,10], and (5) bacteria previously found at increased concentrations after WWTP treatment [16,17]. Some genera include bacteria that match with more than one of the five selection criteria.

To compare the relative abundances of target taxa among locations, we first rarefied all ASV counts to a sequencing depth corresponding to the sample with the lowest coverage (18,059 reads per sample). A pseudo-count of 1 was added to all counts to enable log transformation of read counts where target taxa were absent in a particular sample or location. The relative abundance of target taxa in a given sample was then calculated as the natural log-ratio between taxon read count and the total sample read count. The log ratio of the limit of detection was determined as ln(2)-ln(total no. of reads). Location differences were tested using the χ^2^ statistic (Kruskal–Wallis test). Pairwise differences among locations were assessed using Dunn’s test using PMCMR [34] with p-value corrections using the Benjamin–Hochberg procedure (1995) [35]. The correlation between absolute counts of Klebsiella spp. and Aeromonas spp. with their relative abundances were confirmed using Pearson correlation analysis.

Differential abundance testing across all observed taxa, at the ASV level, was performed with DESeq2 [36] with default settings after applying a variance-stabilizing transformation of the raw counts [37]. We first analyzed differential ASV abundances between clinical (i.e., hospital and nursing home) and domestic wastewater samples. Then, among significantly more abundant ASVs in clinical versus domestic wastewater samples (hereafter referred to as clinically enriched taxa), we assessed whether they were more abundant in influent samples than in domestic wastewaters. Effect sizes and test statistics for pairwise contrasts between (groups of) locations were considered significant at false discovery rate (FDR)-corrected q < 0.1.

#### 2.3.2. Indication of Temporal Effects on Phylum Abundance

Temporal changes in bacterial community composition in water are often driven by temperature [38]. Therefore, the ambient temperature was measured on the day of sampling to relate to temporal patterns of the bacterial community composition in the water samples. Acknowledging microbial community compositionality, the QIIME2-plugin for Songbird [39] was used to run multinomial regression for estimating seasonality in bacterial phylum abundances for each location. Phylum Bacteroidetes was chosen as the reference frame standard for this regression analysis, because Bacteroidetes showed stable abundances in all the locations.

## 3. Results and Discussion

In this study, we analyzed the bacterial compositions along a whole wastewater pathway.

### 3.1. Bacterial Composition Differs between Sources and along the Studied Wastewater Pathway

The phylogenetic structure of wastewater microbial communities differed significantly between sampling locations. Significant differences among the three source locations, as well as WWTP influent and effluent microbial communities are shown in pairwise PERMANOVA analyses (Appendix A). The assumption of homogeneity of multivariate dispersion was met for all but one comparison for weighted UniFrac, but never for unweighted UniFrac. The strong ordination patterns and the large fraction of explained variation (R2 = 39–69%) suggest that significant differences are unlikely to represent statistical artifacts. 

The differences observed between wastewater sources might be caused by the higher use of antibiotics and other drugs in the clinical settings [12]. Only one other study reported a direct comparison between hospital wastewater, domestic wastewater and influent [40], and showed that these three waters were contained in the same cluster, with domestic wastewater more similar to hospital wastewater than to influent. 

In contrast, in our study, domestic wastewater was more similar to influent than to hospital wastewater (Figure 1). Since in Sneek influent is mainly sourced by domestic wastewater from different neighborhoods, it was expected that the bacterial community structure (weighted UniFrac PCoA) of influent was most similar to domestic wastewater. However, the phylogenetic composition (unweighted UniFrac PCoA) of influent resembled nursing home wastewater more than municipal wastewater (Figure 1B), this could be related to the distance of the sampling points to the WWTP, with the nursing home more closely located to the WWTP than the municipal wastewater location (Appendix A). Differences between raw wastewater and influent could result from passage through the sewer pipes from the source to the WWTP, which alters bacterial composition by shifting dominance from obligate anaerobes to facultative anaerobes [41], and probably is determined by the length of the sewer network. Larger group dispersions in weighted versus unweighted UniFrac suggest variation in relative abundances among bacterial lineage groups among samples at each location. Further investigation is necessary to determine whether such variations may be caused by the influence of other wastewater sources and rain events, which fluctuate throughout the year.

While communities from the hospital, nursing home, and domestic wastewater formed separate clusters, they clustered more closely to WWTP influent than to effluent or surface water. This held true both for the relative abundances of taxonomic groups (weighted UniFrac PCoAs, Figure 1A and Appendix A) and their presence/absence (unweighted UniFrac PCoAs, Figure 1B and Appendix A). This is as expected as raw wastewater bacterial communities are formed mainly by human gut bacteria [16,42], while effluent bacterial communities are a mixture of both wastewater and activated sludge bacteria from the WWTP [16,40], and surface water in the WWTP vicinity contains mostly environmental bacteria to which effluent bacteria are added [43,44,45].

### 3.2. Microbial Community Differences between Clinical and Non-Clinical Wastewaters

In order to investigate differences between the raw wastewater sources in more detail, the relative abundance of single bacterial taxa was compared between clinical and domestic wastewater. Significant differences were observed in the relative abundance of 335 bacterial taxa collected from clinical wastewaters (i.e., hospital and nursing home sources) and domestic wastewater. 

The differences between bacterial compositions of clinical and non-clinical wastewaters were examined in just a few studies [14,40,46], and differential community composition of hospital and domestic wastewater did not exhibit a general pattern across studies or locations. Quintela-Baluja et al. [40] reported that hospital wastewater was best characterized by Lactobacillales and Enterobacteriales, while domestic wastewater was best characterized by Clostridiales and Erysipelotrichales. In contrast, our results indicated that a large part of the clinically enriched taxa belonged to Clostridiales (25%, *n* = 51, Appendix A). Our data also showed that only eight of the clinically enriched taxa belonged to Lactobacillales or Enterobacteriales (Appendix A). A likely explanation for the differences in observations resides in the difference in geographical locations at which the studies were conducted, because the human microbiome is country-specific [47]. Other likely explanations are the type of medication used in the different clinical settings [48], and also the methodology of obtaining and processing data - which can all affect the final results. More studies investigating the differences between clinical and non-clinical wastewater microbiomes are therefore necessary to establish which bacteria are typically clinically enriched.

Of the 335 taxa that were found to differ significantly in relative abundances between clinical and non-clinical wastewater, 62% (*n* = 207) were enriched in both hospital and nursing home wastewater (and termed “clinically enriched taxa”, Figure 2). Only 15% of the clinically enriched taxa (*n* = 30) belonged to the target bacteria of relevance listed in Table 2; most of them belonged to the genus *Bacteroides* (*n* = 16) (Appendix A). Five other taxa listed in Table 2—*Aeromonas*, *Enterobacter*, *Klebsiella*, *Pseudomonas*, and *Streptococcus* spp.—were also found between the clinically enriched taxa, but only at very low relative abundance.

### 3.3. Clinical Wastewater Does Not Impact the Overall Bacterial Composition of Influent 

In order to investigate whether the clinically enriched taxa still showed increased abundance in WWTP influent (representing a mix of clinical and domestic wastewater), the abundance of clinically enriched taxa was compared between raw domestic wastewater and WWTP influent. Most of the taxa that have a higher relative abundance in clinical wastewater do not impact the overall bacterial community in influent (Figure 2 and Appendix A). Only 10 out of the 207 clinically enriched taxa were significantly more abundant in influent than in domestic wastewater (Figure 3, Appendix A). In addition, influent clustered more closely to domestic wastewater than to the clinical wastewaters in the weighted UniFrac PCoA ordination (Figure 1A). Seeing as that the overall bacterial composition of influent is most comparable with that of domestic wastewater, our results indicate that clinical wastewater has a limited impact on the abundance of target taxa in WWTP influent for the locations investigated in this study.

The low impact of clinical wastewater on the influent can be explained by the low volume of both hospital and nursing home wastewaters in this study (they constitute less than 1% of the total influent). In two other studies, the bacterial composition of influent that received hospital wastewater was more similar to influent that did not receive hospital wastewater than to the hospital wastewater [14,46], which also indicates that hospital wastewater is too much diluted by other wastewater sources to be traced back in the receiving influent. In our previous study, the impact of hospital wastewater on culturable AMR bacteria in influent was also low [12]. Indeed, considering the dilution factor of the clinical wastewater in influent (1:100), the abundance of bacteria in clinical wastewater has to be 100 times greater than in non-clinical wastewater to affect influent. This was not the case for any of the clinically enriched taxa in this study (Appendix A). 

Among the ten clinically enriched taxa, one species (*Bacteroides stercoris*, which can cause abdominal infections [49]), belongs to the selected target taxa that contain pathogenic bacteria. *Bacteroides* spp. was the only target taxa present in a high relative abundance in both hospital and nursing home wastewater (Appendix A). *Bacteroides* spp. are found to be present in fecal samples taken from AAD patients [8], and are also shown to increase after treatment with fluoroquinolones and β-lactams [9]. In addition, the genus Arcobacter also contains pathogenic species [50], and some species of the genus *Prevotella* are associated with rheumatic diseases [51]. Thus, although clinical wastewater does not impact the overall bacterial composition in influent, it can be a source for some of the pathogenic species found in influent. 

### 3.4. Decrease of Relative Abundance in Most Wastewater Taxa during WWTP Treatment

We observed a decrease of relative abundance in most target taxa (Table 2) after WWTP treatment by at least factor 10 (one log ratio) (Appendix A). Likewise, genera including clinically enriched taxa found in influent were reduced by at least one log-ratio in the WWTP (Appendix A). The decrease in the relative abundance could be due to bacterial removal in the WWTP, or due to the “dilution” of wastewater bacteria by activated sludge bacteria in the effluent. Indeed, the bacterial composition of effluent had the highest diversity among all locations (Appendix A). Thus, one cannot infer a reduction in absolute concentrations from changes in relative abundance. Figure 4 shows the results of the absolute counts of *Klebsiella* spp. and *Aeromonas* spp. from our previous study [12] next to their relative abundances. A similar pattern is observed for both results over the locations, and the correlation is confirmed by Pearson correlation analysis (R = 0.85 and 0.87; Figure 4). In our previous study we showed that *E. coli*, *Klebsiella* spp. and *Aeromonas* spp. were significantly reduced (<99%) in the WWTP [12], therefore, it is most likely that the other genera are also reduced in the WWTP. Although most genera decreased in their relative abundance in the WWTP, all of them were still present in at least 68% of the effluent samples. The number of pathogens surviving could be important from a human health risk assessment standpoint.

*Legionella*, *Mycobacterium*, *Clostridium* and *Leptospira* spp. represent four genera that include pathogenic species, and that show an increased relative abundance from influent to effluent (Appendix A). Other studies have also reported an increase of these four bacteria in the WWTP, as shown in [16,17]. *Clostridium* spp. are common inhabitants of the human gut, and they play an important role in the maintenance of gut homeostasis [52]. Since activated sludge can be a reservoir of *Clostridium* spp. [53], this might explain the relatively high abundance of *Clostridium* spp. found in effluent. Among *Mycobacterium* spp., *M. tuberculosis* complex is an important pathogen. Many environmental nontuberculous mycobacteria can also be pathogenic [54]. *Mycobacterium* also represents a foaming bacterium in activated sludge [55], which could be a reason for the observed increase in the WWTP. *Legionella* spp. are commonly found in moist soil and water, and some *Legionella* species can cause community and hospital-acquired pneumonia [56]. In the past years, some industrial WWTPs in the Netherlands have shown to be a source for pneumonia caused by *Legionella* bacteria [57]. *Leptospira* spp., causing leptospirosis, which is associated with rainfall and flooding, can persist for several months in the environment without a host [58], which might explain why it persists in the WWTP as well. Pathogenic species belonging to these four genera might be problematic for public health if released into surface water in sufficient quantities. However, it should be mentioned that these findings are based on the genus level, and various non-pathogenic species belong to these four genera as well. Further research, i.e., by using whole genome sequencing techniques can provide more insight about the exact species included in these genera.

### 3.5. The Bacterial Composition Throughout the Year Only Differs in Surface Waters

Changes in bacterial phylum composition potentially caused by temperature changes were limited to surface water samples (Figure 5). For both the human gut phylum of Firmicutes and the environmental bacteria Cyanobacteria, the relative abundance did not significantly change in wastewater and effluent upon fluctuations in temperature, which is in concordance with a recent similar study [46]. The wastewater sources and influent showed a stable bacterial community structure. These waters are mainly dominated by human gut bacteria [16], which in healthy individuals is a stable community [59]. The stable bacterial community structure of effluent can be explained as effluent is mostly sourced by influent and activated sludge bacteria [16,40], which consists of a large part of core bacteria which are present all year round [60]. 

In contrast, temperature affected abundances in surface water. Still, this effect is limited to Cyanobacteria (up-, downstream, and control surface water, *p* < 0.001) and Planctomycetes (up- and downstream surface water, *p* < 0.001), in agreement with the role of temperature- and light intensity driven cyanobacterial growth [61]. In another study, temperature also had no substantial impact on the composition at the Phylum level in river water [62]. The reason why the influence of temperature was limited to Cyanobacteria and Planctomycetes can be explained as Cyanobacteria are photoautotrophic and therefore have a benefit to other bacteria. Higher densities of Planctomycetes are reported after cyanobacterial blooms, suggesting a possible association of this phylum with Cyanobacteria [63]. 

### 3.6. Culturing More Sensitive Than Sequencing to Detect Impact of This WWTP on Surface Water

In our previous work on the same water samples, an increase in the concentrations of *E. coli* was found in the surface water downstream from the WWTP in comparison to the WWTP upstream location and the control location, as determined by culture [12]. 16S rRNA gene amplicon sequencing was not able to detect any impact of the studied WWTP on the receiving surface water. Although most of the investigated genera had a decreased relative abundance in the WWTP, they were still present in at least 68% of the effluent samples, which are then released into the receiving surface water. However, their relative abundances did not differ between the upstream, downstream, and control surface water samples (Figure 4 and Appendix A). In addition, up- and downstream samples clustered together in the UniFrac plots and did not significantly differ (Appendix A). 

In the receiving surface water, effluent bacteria are mixed with environmental bacteria. Therefore, the relative abundance of effluent bacteria is reduced, making it more difficult to detect differences in effluent bacteria between surface waters up- and downstream from the WWTP. In a similar study, the difference in bacterial community structure between up and downstream water was also not significant [46]. In another comparable study, a difference between up- and downstream water bacterial composition was observed. However, in this study, the sampling was performed in summer when dilution of effluent in the receiving water was minimized [40]. Overall, the influence of WWTP effluent may differ per WWTP and depend on its degree of dilution with surface water. Methods other than 16S rRNA gene amplicon sequencing, such as bacterial enumeration of taxa present predominantly in effluent by culture, may be necessary to detect WWTP effects in situations when effluent is highly diluted.

This study provides insight in the microbial community structure in wastewaters obtained from different sources. It is the first study to highlight the differences in community structure between clinical and non-clinical wastewaters. Nevertheless, some limitations should be noted. The 16S rRNA gene amplicon sequencing does not allow for the strain-level resolution needed to specifically detect pathogenic (sub-) species. Thus, this study was primarily directed at generating hypotheses about pathogens that could potentially be enriched in clinical wastewaters. Follow-up studies applying different methods than 16S rRNA profiling are therefore needed to verify whether the pathogenic or potentially target taxa are indeed not enriched in clinical wastewaters. The 16S, furthermore, only provides information about relative abundances instead of absolute concentrations. However, comparisons with our previous study from which we obtained absolute counts of *Klebsiella* spp., and *Aeromonas* spp., showed a similar pattern of absolute concentrations and relative abundances along the wastewater pathway. Still, detection by absolute means (i.e., culturing or quantitative PCR (qPCR)) would be needed to confirm the decline observed in the WWTP treatment. Furthermore, the results of this study are limited to the particular characteristics of the sampled locations, i.e., the volume of wastewater originating from the sampled hospital and nursing home relative to the total community, the degree of use of medicines in the hospital and nursing home, the properties of the WWTP, and the dilution factor of WWTP effluent in the receiving surface water. Future comparisons between multiple studies will help elucidate the range that the impact of single healthcare institutions on the overall municipal wastewater can have.

## 4. Conclusions

In summary, this study provides new insights into shifts in bacterial community composition from wastewater sources to the environment. We found that clinical and non-clinical wastewaters significantly differ in their composition, but this difference was mainly caused by genera not included within the target taxa.

Clinical wastewater had little impact on the bacterial composition found in influent. From the 207 clinically enriched taxa, only 10 were more abundant in influent than in domestic wastewater; however, a part of these consisted of genera also included pathogenic genera. Most of the bacterial genera investigated in this study decreased in their relative abundance in the WWTP, except for *Clostridium* spp., *Mycobacterium* spp., *Legionella* spp., and *Leptospira* spp. Still, all taxa studied were detected in the majority of the effluent samples. For an assessment of the impact on public health, absolute concentrations of confirmed pathogens would be needed. 

In this study, 16S rRNA gene amplicon sequencing was not sensitive enough to demonstrate the impact of the WWTP on the surface water, in contrast to previous culture-based analyses. Finally, in this study, temperature had a limited impact on the bacterial composition in surface water, and the impact on the bacterial composition in wastewater and effluent was negligible. Therefore, sampling campaigns studying microbial communities in wastewater might not necessarily have to cover all seasons. In conclusion, our results suggest a limited role of clinical wastewaters on the bacterial community in the receiving treatment plant. Furthermore, both culture- and DNA-based analyses should be combined to better elucidate the impact of WWTP effluents on the environment. 

## Figures and Tables

**Figure 1 microorganisms-09-00718-f001:**
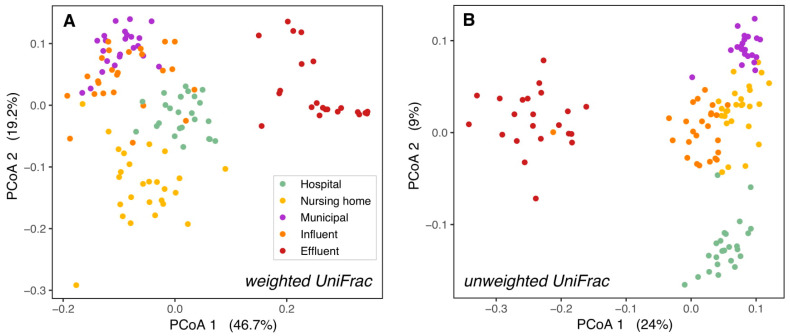
Bacterial ß-diversity along the wastewater pathway. Principal coordinates ordination of different wastewaters based on weighted UniFrac distances (**A**) and unweighted UniFrac distances (**B**). 66% of the diversity in bacterial composition and 39% of the diversity in bacterial membership could be explained by the locations. The percentage of variation explained by each axis is shown between parentheses. H = hospital, N = nursing home, M = municipal, I = influent, E = effluent.

**Figure 2 microorganisms-09-00718-f002:**
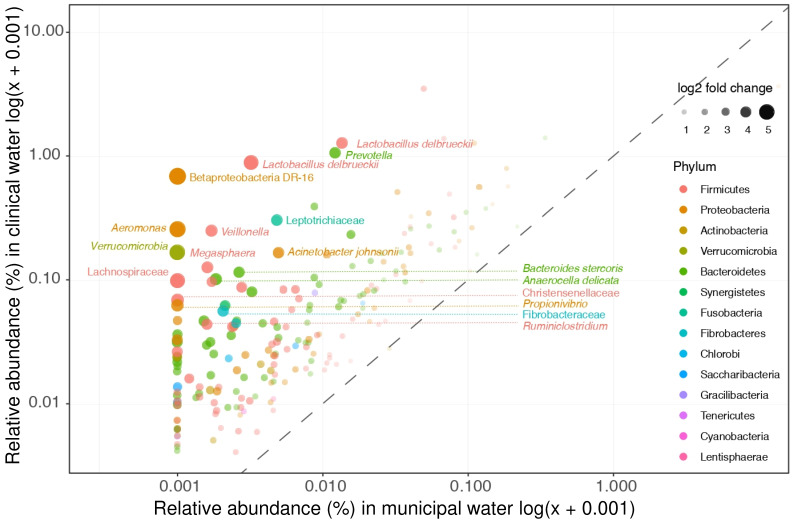
Clinical enriched taxa. Clinical enriched taxa were defined as taxa that were significantly more abundant in both hospital and nursing home wastewater as compared to municipal wastewater. The relative abundances of these taxa in hospital and municipal wastewaters are depicted. The clinical most enriched taxa are not members of our selected target taxa (i.e., *Lactobacillus delbrueckii* and *Prevotella*).

**Figure 3 microorganisms-09-00718-f003:**
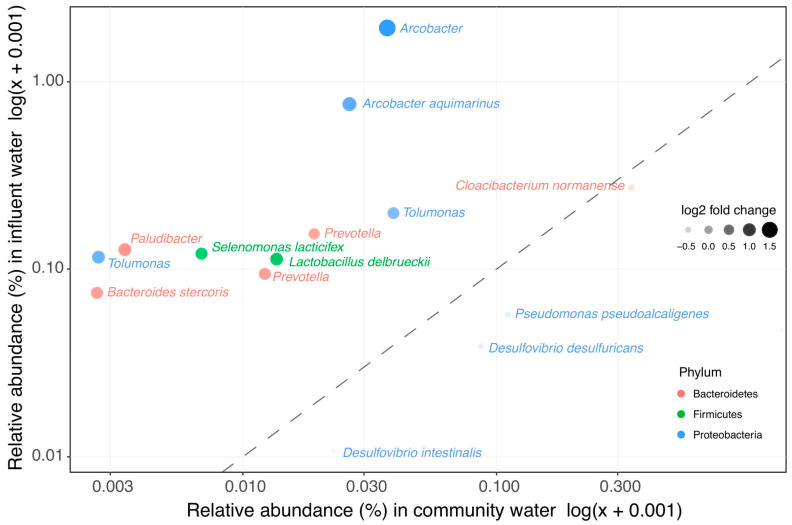
Clinical enriched taxa in influent vs. wastewater collected from households. The 207 taxa found to be enriched in clinical wastewaters were compared in their relative abundance in influent and in the wastewater collected from households. In total, 10 taxa were significantly more abundant in influent than in wastewater collected from households.

**Figure 4 microorganisms-09-00718-f004:**
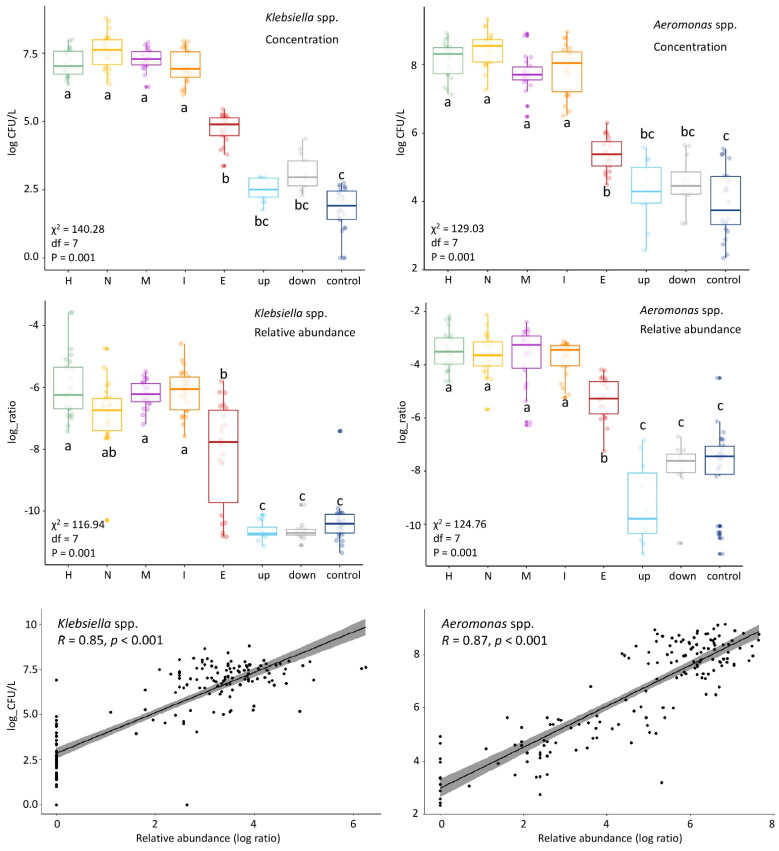
Comparison of colony forming units (CFU) counts and relative abundances in the different locations of *Klebsiella* spp. (left) and *Aeromonas* spp. (right). The CFU counts (log(CFU/L) and relative abundance (log ratio) show similar patterns along the locations. This was confirmed by Pearson correlation analysis (R) shown in the linear regression curves. Kruskal-Wallis statistics of the CFU counts and relative abundances are shown on the left side of each panel: chi-squared (χ^2^), df and p-value). Group differences were assessed by the Dunn’s test with the p-value adjustment method: BH. The locations are shown on the horizontal axis. H = hospital, N = nursing home, C = community (municipal wastewater), I = influent, E = effluent, up = upstream surface water, down = downstream surface water, and control = surface water collected at nature reserve “de Deelen”.

**Figure 5 microorganisms-09-00718-f005:**
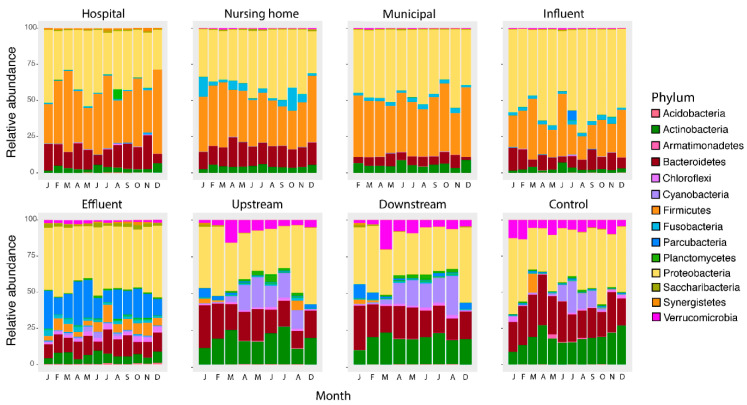
Relative abundance at the Phylum level per location per month. The main groups in wastewaters (hospital, nursing home, municipal wastewater and influent) consist of Proteobacteria, Firmicutes, Bacteroidetes and Actinobacter, which are significantly changed after the wastewater treatment with Proteobacteria and Parcubacteria as main groups in the effluent. Surface waters are similar to each other but form a distinct group dominated by Proteobacteria, followed by Actinobacteria and Bacteroidetes as the main Phyla.

**Table 1 microorganisms-09-00718-t001:** 16S rRNA gene amplicon sequencing results. The number of samples obtained per location, as well as the number of reads (average, minimal and maximal) are shown together with the rarefied and non-rarefied amplicon sequence variants (ASV) and the resulting average detection limit at log ratio. Libraries were rarefied for statistical comparison to 18,059 reads per sample.

Location	Number of Samples Sequenced	Average Number of Reads	Minimal Number of Reads	Maximal Number of Reads	Non-Rarefied ASV Count	Rarefied (to 18059) ASV Count	Average Detection Limit ^1^ (Log Ratio)
Hospital	25	55,638	28,811	138,846	4920	4185	−10.88
Nursing home	26	52,985	29,657	104,195	3913	3386	−10.83
Community	23	56,871	31,794	101,117	3649	3156	−10.91
Influent	25	63,326	34,958	97,559	6069	4932	−11.01
Effluent	22	50,806	25,594	77,211	8499	7469	−10.79
Upstream	12	44,054	25,061	68,055	4942	4359	−10.65
Downstream	12	44,281	18,059	66,764	4980	4404	−10.66
Control	26	37,912	20,363	85,529	6565	5851	−10.47
Average	21	50,734	26,787	92,410	5442	4718	−10.78

^1^ The log ratio of the detection limit per sample is calculated by: log(1)—log(total number of reads).

**Table 2 microorganisms-09-00718-t002:** Target taxa. Taxa that include pathogenic bacteria of clinical interest monitored from wastewater to environment. If no reference is given in the cell, the bacteria does not fall under this category.

Genus/Species	ESKAPE Pathogen	WHO Priority Pathogen for AMR ^1^	AAD Associated/Increased after AB Treatment	Water-Based Pathogen	WWTP Increase
*Escherichia* spp./*Shigella* spp. ^2^		[4] (a/c)			
*Klebsiella* spp.	[2]	[4] (a)	[12]		
*Enterobacter* spp.	[2,1]	[4] (a)			
*Proteus* spp.		[4] (a)			
*Serratia* spp.		[4] (a)			
*Providencia* spp.		[4] (a)			
*Morganella* spp.		[4] (a)			
*Salmonella* spp.		[4] (b)			
*Mycobacterium* spp.		[4] (a)		[23,25]	[15]
*Enterococcus* spp.	[2]	[4] (b)			
*Bacteroides* spp.			[13,14]		
*Acinetobacter baumanii*	[2]	[4] (a)			
*Pseudomonas aeruginosa*	[2]	[4] (a)		[25]	
*Staphylococcus aureus*	[2]	[4] (b)	[12]		
*Helicobacter pylori*		[4] (b)		[24]	
*Campylobacter* spp.		[4] (b)			
*Neisseria gonorrhoeae*		[4] (b)			
*Streptococcus pneumoniae*		[4] (c)			
*Haemophilus influenzae*		[4] (c)			
*Clostridium* spp.			[12,14]		[15]
*Aeromonas* spp.				[24]	
*Legionella* spp.				[25]	[16]
*Leptospira* spp.				[26]	[16]
*Vibrio* spp.				[26]	

^1^ WHO priority pathogens for AMR are divided into three categories: (a) Critical, (b) High, and (c) Medium. ^2^
*Escherichia* spp. and *Shigella* spp. could not be distinguished from one to the other.

## Data Availability

Sequence data is available in the NCBI sequence read archive (SRA) under project numbers PRJNA668059 and PRJNA668064.

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
