# Peer review of "Effects of Clinical Wastewater on the Bacterial Community Structure from Sewage to the Environment"

_microorganisms, 2021, doi:10.3390/microorganisms9040718_

Round 1

Reviewer 1 Report

Manuscript number: microorganisms-1150659

Effects of clinical wastewater on the bacterial community structure from sewage to the environment

The study aimed to investigate the contribution of clinical wastewater to the changes in bacterial composition in wastewater treatment line. Specifically, both spatial and temporal variations of bacterial community were taken into account by a monthly sampling campaign in seven different locations from sewage to the environment. The study showed interesting and significant results, making the manuscript worth to be published. However, some polishing works still needed to improve the readability and comprehension of the manuscript as follows.

  • There should be a substantial improvement of the abstract as its current version is not clear and un qualified. Specifically, the first sentence is intricate with vague meaning. There is no explanation of the methodology of the study and the goal(s) and the results of the study are not clear.
  • No keywords can be found
  • Line 64-89, there is a too detailed description of the methodology.
  • Line 93-94: It should change to “More details of the sampling campaign can be found in ...”
  • Line 134: It should be 2.3.
  • Line 227-229: Is there any evidence to support this implication?
  • Figures 1 and 2: The axis titles need to be placed outside of the graphs.
  • Figures 1 and 3: The background should be transparent like the other figures.
  • Line 396: it should be 3.6
  • Line 443-446: it should be “… was mainly… that excluded…”
  • Line 451-452: These bacteria should be in Italic
  • Line 458: The impact of what?
  • The title of the subsections in section 3 should be concise.

Author Response

The authors would like to thank the reviewer for his kind words, and also for the time and effort to read and improve our manuscript. The comments are processed as follows:

  • There should be a substantial improvement of the abstract as its current version is not clear and un qualified. Specifically, the first sentence is intricate with vague meaning. There is no explanation of the methodology of the study and the goal(s) and the results of the study are not clear.

We agree with the reviewer comments. We partly rewrote the abstract to make it more clear and to include the goal and methodology. The new abstract is:

“This study pertains to measure differences in bacterial communities along the wastewater pathway, from sewage sources through the environment. Our main focus was on taxa which include pathogenic genera, and genera harbouring antibiotic resistance (henceforth referred to as “target taxa”). Our objective was to measure the relative abundance of these taxa in clinical wastewaters compared to non-clinical wastewaters, and to investigate what changes can be detected along the wastewater pathway. The study entailed a monthly sampling campaign along a wastewater pathway, and taxa identification through 16S rRNA amplicon sequencing. Results indicated that clinical and non-clinical wastewaters differed in their overall bacterial composition, but that target taxa were not enriched in clinical wastewater. This suggests that treatment of clinical wastewater before release into the wastewater system would only remove a minor part of the potential total pathogen load in wastewater treatment plants. Additional findings were that the relative abundance of most target taxa was decreased after wastewater treatment, yet all investigated taxa were detected in 68% of the treated effluent samples – meaning that these bacteria are continuously released into the receiving surface water. Temporal variation was only observed for specific taxa in surface water, but not in wastewater samples.”

  • No keywords can be found

The keywords were indeed not provided. We added the following keywords: 16S rRNA amplicon sequencing; bacterial community structure; clinical wastewater; sampling campaign; wastewater pathway.

  • Line 64-89, there is a too detailed description of the methodology.

We agree with the reviewer’s comment. Therefore, we narrowed this part of the introduction down to:

“This study aimed to investigate the contribution of clinical wastewater on the bacterial composition in wastewater and the environment. We used 16S rRNA to broadly screen the relative abundance and fate of bacterial target taxa that were selected based on pathogenic potential: i) pathogens of clinical relevance and AMR features (ESKAPE pathogens and WHO priority pathogens) [2-4], ii) water-based pathogens that can grow and thrive in water systems [18-22], iii) antibiotic-associated diarrhoea (AAD) bacteria [8-10] and, iv) bacteria that increase after WWTP treatment [16, 17] (Table 2).

The results of this study provide insight into the community composition origin in a wastewater chain, the difference between clinical and non-clinical wastewater, and how this affects the bacterial community in the WWTP and the receiving surface water. In doing so, the study sheds light on the likely fate of potentially pathogenic bacteria in this wastewater chain.”

  • Line 93-94: It should change to “More details of the sampling campaign can be found in ...”

We rewrote the whole sentence: “While a more detailed description of the sampling campaign can be found in our previous research [12] a brief summary of the study is as follows:”

  • Line 134: It should be 2.3.

This is correct. We changed 2.2 into 2.3 and also changed the numbers of the subsections (2.3.1 and 2.3.2)

  • Line 227-229: Is there any evidence to support this implication?

We could not find evidence to support this implication, therefore we changed the sentence: “Further investigation is necessary to determine whether such variations may be caused by the influence of other wastewater sources and rain events, which fluctuates throughout the year.”

  • Figures 1 and 2: The axis titles need to be placed outside of the graphs.
  • Figures 1 and 3: The background should be transparent like the other figures.

We agree that Figures 1-3 could be improved. We made the following changes: we placed the axis titles outside of Figures 1 and 2. We placed the legend of Figure 1 within the figure. We made the background of Figure 1 and 3 white. We made the bacterial names in Figure 3 in italics and readable within the figure. Finally, we stored the figures with higher quality.

  • Line 396: it should be 3.6

This is correct. We changed 3.1 into 3.6.

  • Line 443-446: it should be “… was mainly… that excluded…”

We agree with this comment and changed it, the sentence is now:

“We found that clinical and non-clinical wastewaters significantly differ in their composition, but this difference was mainly caused by genera not included within the target taxa.”

  • Line 451-452: These bacteria should be in Italic

We changed the style of the bacterial names into italic in these lines, and checked the rest of tha manuscript for this.

  • Line 458: The impact of what?

We were referring to the bacterial composition, to make this clear we added it to the sentence, the sentence is now:

“Finally, in this study temperature had a limited impact on the bacterial composition in surface water, and the impact on the bacterial composition in wastewater and effluent was neglectable.”

  • The title of the subsections in section 3 should be concise.

We adjusted the titles to make them fit on one line.

Reviewer 2 Report

A review of the manuscript ID: microorganisms-1150659, entitled “Effects of clinical wastewater on the bacterial community structure from sewage to the environment”

Authors: Ilse Verburg et al.

Overall, the manuscript is interesting and shows the important issues. Therefore, in my opinion, it is worth publishing in the Microorganisms journal. However, authors need to improve the current version of the manuscript.

My comments and suggestions:

1) please complete the keywords

2) pg 1, ln 37-38: commas - remove italics

3) the introduction section must be modified and improved, especially the last paragraphs, which are not appropriate for introduction

4) Fig. 2 – not readable, please improve quality

5) Fig. 3 – not readable, please improve quality and Latin names must be italicized

6) Fig. 4 – please improve quality

7) please read the manuscript carefully and correct all minor deficiencies

Author Response

The authors would like to thank the reviewer for his kind words, and also for the time and effort to read and improve our manuscript. The comments are processed as follows:

  • please complete the keywords

The keywords were indeed not provided. We added the following keywords:

16S rRNA amplicon sequencing; bacterial community structure; clinical wastewater; sampling campaign; wastewater pathway.

  • pg 1, ln 37-38: commas - remove italics

We changed the format, the commas are not in italics now.

  • the introduction section must be modified and improved, especially the last paragraphs, which are not appropriate for introduction

We agree with the reviewer’s comment. Therefore, we narrowed the last part of the introduction (lines 64-89) down to:

“This study aimed to investigate the contribution of clinical wastewater on the bacterial composition in wastewater and the environment. We used 16S rRNA to broadly screen the relative abundance and fate of bacterial target taxa that were selected based on pathogenic potential: i) pathogens of clinical relevance and AMR features (ESKAPE pathogens and WHO priority pathogens) [2-4], ii) water-based pathogens that can grow and thrive in water systems [18-22], iii) antibiotic-associated diarrhoea (AAD) bacteria [8-10] and, iv) bacteria that increase after WWTP treatment [16, 17] (Table 2).

The results of this study provide insight into the community composition origin in a wastewater chain, the difference between clinical and non-clinical wastewater, and how this affects the bacterial community in the WWTP and the receiving surface water. In doing so, the study sheds light on the likely fate of potentially pathogenic bacteria in this wastewater chain.”

  • 2 – not readable, please improve quality

We placed the axis titles outside of the figure and stored the figure with higher quality.

  • 3 – not readable, please improve quality and Latin names must be italicized

We agree that Figure 3 could be improved. We made the background white and changed the bacterial names in Figure 3 in italics and made them readable within the figure. Finally, we stored the figure with higher quality.

  • 4 – please improve quality

We improved the quality of the figure and enlarged some of the symbols to make it more readable.

  • please read the manuscript carefully and correct all minor deficiencies

The manuscript has been checked with an English proofreading and all the detected deficiencies have been improved.

Reviewer 3 Report

The manuscript “Effects of clinical wastewater on the bacterial community structure from sewage to the environment” (Microorganisms-1150659) is well written and presented. The study compared microbial communities using 16S rRNA amplicon sequencing in clinical and non-clinical wastewater, influent, effluent, upstream, downstream and control samples over one year, and specified target taxa related to pathogens and AMR. The statistical analyses applied were appropriate, and the discussion provided in-depth knowledge. It is appreciated that the discussion included both findings and limitations of the study. I have a few minor comments detailed below.

  1. Please provide keywords
  2. Line 51, “…previous studies showed contradictory results…” please give some details of these contradictory results.
  3. In section 2.1, it could be more informative if the authors could provide the distances of the sampling points to the WWTP, if available.
  4. Line 116, please give the primers that amplify the V3-V4 regions of the 16S rRNA gene.
  5. In results section 3.1, Line 224, what would be the possible reasons that using the unweighted UniFrac distance resulted that influent and nursing home wastewater are similar? Could this be related to the distance of sampling points (see my comment 3)? Some studies have shown that the influent is large affected by the indigenous sewer microbiome, therefore sewerage length may be a factor. Or other factors, if the author may speculate one or two?
  6. Line 351, referring to Figure 4 seems not linked to the context.
  7. Line 396, the section numbering should be corrected to “3.7” instead of “3.1”.

Author Response

The authors would like to thank the reviewer for his kind words, and also for the time and effort to read and improve our manuscript. The comments are processed as follows:

  1. Please provide keywords

The keywords were indeed not provided. We added the following keywords: 16S rRNA amplicon sequencing; bacterial community structure; clinical wastewater; sampling campaign; wastewater pathway.

  1. Line 51, “…previous studies showed contradictory results…” please give some details of these contradictory results.

To provide more details of the contradictory results about the impact of hospital wastewater on the receiving WWTP, we replaced the sentence:

“Previous studies showed contradictory results about the impact of clinical wastewater in influent [13, 14].”

with the following sentence:

“Paulus et. al. showed that pre-treatment of hospital wastewater helps to reduce the presence of ARGs in the receiving WWTP [13], yet Buelow et. al. did not observe differences in the relative abundance of ARGs in WWTPs that did, or did not receive hospital wastewater [14].”

  1. In section 2.1, it could be more informative if the authors could provide the distances of the sampling points to the WWTP, if available.

This is a good suggestion from the reviewer. We do not have information about the exact distances of the sampling points to the WWTP. However, we made an extra supplementary figure which shows a map in which the sampling locations are indicated. We refer to this figure in section 2.1 as “Supplementary Figure S1”, and changed all the other supplementary figure numbers accordingly. 

  1. Line 116, please give the primers that amplify the V3-V4 regions of the 16S rRNA gene.

We added information about the primers used to amplify the V3-V4 regions. We changed

“The V3-V4 regions of the 16S rRNA gene 119 were amplified using Amplicon primers with overhang adaptors, followed by a second 120 PCR reaction attaching indices and Illumina sequencing adapters using the Nextera®XT 121 Index Kit (Illumina, USA)”

into the following:

“The V3-V4 regions of the 16S rRNA gene were amplified by the polymerase chain reaction (PCR) using Amplicon primers with overhang adaptors (16S Amplicon PCR Forward Primer = 5' TCGTCGGCAGCGTCAGATGTGTATAAGAGACAGCCTACGGGNGGCWGCAG 16S Amplicon PCR Reverse Primer = 5' GTCTCGTGGGCTCGGAGATGTGTATAAGAGACAGGACTACHVGGGTATCTAATCC). A second PCR was used for attaching indices and Illumina sequencing adapters using the Nextera®XT Index Kit (Illumina, USA)”

  1. In results section 3.1, Line 224, what would be the possible reasons that using the unweighted UniFrac distance resulted that influent and nursing home wastewater are similar? Could this be related to the distance of sampling points (see my comment 3)? Some studies have shown that the influent is large affected by the indigenous sewer microbiome, therefore sewerage length may be a factor. Or other factors, if the author may speculate one or two?

The nursing home is more closely located to the WWTP than the residential area from which we obtained the municipal wastewater sample. We partly rewrote this part of section 3.1. in which we mention the probability of the distance to the WWTP causing differences observed between raw wastewater and WWTP influent water and refer to the supplementary figure showing the distances of the sampling locations to the WWTP (see comment 3).

  1. Line 351, referring to Figure 4 seems not linked to the context.

The reviewer is right, referring to Figure 4 is unsuitable in this context, we removed “Figure 4” and referred only to “Supplementary Figure S2”.

  1. Line 396, the section numbering should be corrected to “3.7” instead of “3.1”.

This section numbering was indeed not correct. We checked the numbering, and changed “3.1” into “3.6” and checked the rest of the document. We found a similar inaccuracy in section 2.3 and corrected this as well.

Reviewer 4 Report

This is an interesting paper that can be published after revision.

The following issues must be addressed:

  1. The Introduction part should contain a sentence related with PhACs removal from wastewater.
  2. The authors must clarify which are the origins of the statistical artifacts.
  3. Explain in more details why the surface water temperature has a low impact on the bacteria abundance.
  4. Overall, the paper need more coherence between results. Some results seems to have no correlation with the others. 

Author Response

The authors would like to thank the reviewer for his kind words, and also for the time and effort to read and improve our manuscript. The comments are processed as follows:

  1. The Introduction part should contain a sentence related with PhACs removal from wastewater.

We added the following sentence to the introduction: “WWTP’s are designed to reduce the biological oxygen demand, nitrogen, phosphorus, and the total suspended solids, but not for the removal of pharmaceutically active compounds and bacteria.”

  1. The authors must clarify which are the origins of the statistical artifacts.

Indeed, we address statistical limitations in line 438-440 and 444-446, to address the limitations of relative vs absolute determination of bacteria.

  1. Explain in more details why the surface water temperature has a low impact on the bacteria abundance.

We added the following sentence to explain why surface water temperature was limited to only Cyanobacteria and Planctomycetes:

“The reason why the influence of temperature was limited to Cyanobacteria and Planctomycetes can be explained as Cyanobacteria are photoautotrophic and therefore have a benefit to other bacteria. Higher densities of Planctomycetes are reported after cyanobacterial blooms, suggesting a possible association of this phylum with Cyanobacteria [63].”

  1. Overall, the paper need more coherence between results. Some results seems to have no correlation with the others. 

 The discussion of each independent step of the wastewater pathway might appear independent, however it follows the approach of analysing a complete wastewater pathway. We attempt to lay this down in the introduction (line 76-78). We added an additional sentence at the beginning of the results and discussion section to clarify this approach.

Reviewer 5 Report

The manuscript reports interesting results, in line with the scope of journal, in particular offers some unique data about effects of clinical wastewater on the bacterial community structure from sewage to the environment.

This paper is composed of Abstract, Introduction, Materials and methods, Results, Discussion and Conclusions. The presented material is very interesting and current. The selection of "Materials and methods" is correct. The manuscript is to the point and its arrangement is correct and transparent. The description of tables and figures is legible and understandable. The authors did not make substantial mistakes.
The use of bibliography is correct.

Other weaknesses to be corrected:
Complete the keywords, keywords should be in alphabetical order.
Explain acronyms and symbols in the text e.g. qPCR,
Please improve the manuscript with a English proofreading

I recommend this manuscript for publication in "Microorganisms" Journal

Author Response

The authors would like to thank the reviewer for his kind words, and also for the time and effort to read and improve our manuscript. The comments are processed as follows:

Complete the keywords, keywords should be in alphabetical order.

The keywords were indeed not provided. We added the following keywords:

16S rRNA amplicon sequencing; bacterial community structure; clinical wastewater; sampling campaign; wastewater pathway.

Explain acronyms and symbols in the text e.g. qPCR,

Some of the acronyms and symbols used were unexplained. We added explanations the first time acronyms were used: amplicon sequence variants (ASV, description Table 1), false discovery rate (FDR, section 2.3.1), polymerase chain reaction (PCR, section 2.2), colony forming units (CFU, description Figure 4), chi-squared (χ2, Figure 4), and quantitative PCR (qPCR , discussion).

Please improve the manuscript with a English proofreading

The manuscript has been checked with an English proofreading and all the detected deficiencies have been improved.

Round 2

Reviewer 1 Report

The authors have revised their manuscript according to my previous comments. These modifications are sufficient and the current manuscript has a high scientific quality and is ready for being published.

Reviewer 2 Report

In my opinion, the current version of the manuscript is much better than the original version. The authors improved the MS according to my suggestions.

Good job.

Congratulations. 

Reviewer 3 Report

The revised manuscript can be accepted for publication. 

Reviewer 4 Report

The authors made the necessary changes. The manuscript can be published in the present form.

Reviewer 5 Report

English language and style are fine. The manuscript is significantly 
improved and now warrants publication in "Microorganisms" Journal.